# RAW: A Robust and Agile Plug-and-Play Watermark Framework for AI-Generated Images with Provable Guarantees

**Xun Xian**
Department of ECE
University of Minnesota
xian0044@umn.edu

**Ganghua Wang**
School of Statistics
University of Minnesota
wang9019@umn.edu

**Xuan Bi**
Carlson School of Management
University of Minnesota
xbi@umn.edu

**Jayanth Srinivasa**
Cisco Research
jasriniv@cisco.com

**Ashish Kundu**
Cisco Research
ashkundu@cisco.com

**Mingyi Hong**
Department of ECE
University of Minnesota
mhong@umn.edu

**Jie Ding**
School of Statistics
University of Minnesota
dingj@umn.edu

## Abstract

Safeguarding intellectual property and preventing potential misuse of AI-generated images are of paramount importance. This paper introduces a robust and agile plug-and-play watermark detection framework, referred to as RAW. As a departure from existing encoder-decoder methods, which incorporate fixed binary codes as watermarks within latent representations, our approach introduces learnable watermarks directly into the original image data. Subsequently, we employ a classifier that is jointly trained with the watermark to detect the presence of the watermark. The proposed framework is compatible with various generative architectures and supports on-the-fly watermark injection after training. By incorporating state-of-the-art smoothing techniques, we show that the framework also provides provable guarantees regarding the false positive rate for misclassifying a watermarked image, even in the presence of adversarial attacks targeting watermark removal. Experiments on a diverse range of images generated by state-of-the-art diffusion models demonstrate substantially improved watermark encoding speed and watermark detection performance, under adversarial attacks, while maintaining image quality. Our code is publicly available here.

## 1 Introduction

In recent years, Generative Artificial Intelligence, notably in computer vision, has made significant strides. The adoption of diffusion models (DM) in applications like Stable Diffusion [1] and DALLE-2 [2] has greatly improved image generation quality. However, these advancements also raise concerns about potential misuse, seen in instances such as DeepFake [3] and copyright infringement [4].

To mitigate the potential misuse of diffusion models, the incorporation of watermarks emerges as a promising solution. Watermarked images, tagged with crafted signals, act as markers to identify their machine-generated origin. Watermarking techniques designed for generative models can be generally

38th Conference on Neural Information Processing Systems (NeurIPS 2024).

classified into two categories: model-specific [5, 6, 7, 8] and model-agnostic [9, 10, 11], as outlined in Table 1. Model-specific methods refer to those tailored for particular generative models, often offering a better tradeoff between watermarked image quality and watermark detection performance. However, this characteristic may potentially restrict their applicability across diverse use cases. For instance, the Tree-Ring watermark [8] is tailored for specific samplers, e.g., DDIM [12], used for image generation within diffusion models. The feasibility of adapting this method to other commonly used samplers remains open to discussion.

Table 1: Summary of features of several representative watermark techniques. The second column denotes the method's suitability for real-time on-the-fly implementation. **/** denotes cases where the watermark is embedded during the generative process. The third column evaluates whether the watermarking method provides provable guarantees on false-positive rates (FPRs) under adversarial attacks in a distribution-free manner.

| Method | Model agnostic | On-the-fly deployment | Provable FPRs under adversarial attacks |
|---|---|---|---|
| DwtDctSvd [13] | ✓ | ✓ | ✗ |
| RivaGan [10] | ✓ | ✗ | ✗ |
| StegaStamp [11] | ✓ | ✗ | ✗ |
| Stable Signature [5] | ✗ | / | ✗ |
| Tree Ring [8] | ✗ | / | ✗ |
| RAW (Our method) | ✓ | ✓ | ✓ |

In contrast, model-agnostic approaches directly watermark generated content without modifying the generative models. These approaches can be categorized into two types. The first, from traditional signal processing, e.g., DwTDcTSvd [13], embeds watermarks in specific parts of images' frequency domains. However, they can be vulnerable to strong image manipulations and adversarial attacks for removing watermarks [14]. The second type employs deep learning techniques, utilizing encoder-decoder structures to embed watermarks, such as binary codes, in latent spaces. For instance, RivaGan [10] trains the watermark and watermark decoder jointly as learned models, enhancing transmission and robustness. However, these methods demand greater computational resources for watermark injection, limiting real-time on-the-fly deployment. For example, when injecting watermarks into images, the RivaGan requires over $15\times$ the time needed by the DwtDct methods [15].

Additionally, there has been a sustained emphasis on precisely evaluating false-positive rates (FPRs) and/or the Area Under the Receiver Operating Characteristic curve (AUROC) for each utilized watermarking strategy [16], given the potential economic implications associated with watermark implementation. To establish an explicit theoretical formulation for FPRs, many studies have assumed that the binary watermark code extracted from unwatermarked images exhibits a pattern where each digit is an independent and identically distributed (IID) Bernoulli random variable with equal probabilities. This assumption enables the explicit derivation of the FPRs when comparing the extracted binary code with the predefined actual binary watermark code. However, such an assumption may not hold as empirically observed in [17], and thus the corresponding formulation for FPRs might be incorrect.

## 1.1 Contributions

In this paper, we introduce a **R**obust, **A**gile plug-and-play **W**atermark framework, abbreviated as **RAW**. RAW is designed for both adaptability and computational efficiency, providing a model-agnostic approach for real-time, on-the-fly deployment of image watermarking (which can be straight-forwardly extended to video watermarking as outlined in Section 4.5). This dedication to adaptability is to ensure the accessibility for third-party users, e.g., artists and generative model providers. More-over, this adaptability is fortified by the integration of state-of-the-art smoothing techniques for achieving provable guarantees on the FPRs for detection, even under adversarial attacks.

**A new framework for robust and agile plug-and-play image watermark learning** In contrast to encoder-decoder techniques that insert fixed binary watermarks into latent spaces, we propose to embed learnable watermarks, matching the image dimensions, into both the frequency and spatial domains of the original images. To differentiate between watermarked and unwatermarked samples, we utilize a classifier, e.g., a convolutional neural network (CNN), and perform joint training for both the watermarks and the classifier. The proposed framework offers several benefits, including enhanced computational efficiency achieved through batch processing for watermark injection post joint training. For instance, our experimental results demonstrate time efficiency enhancements, approximately $30\times$

($200\times$) faster than the frequency-based (encoder-decoder based) method, respectively. Moreover, it can be readily integrated with other state-of-the-art techniques to further enhance robustness and generalizability, such as adversarial training [18, 19], and contrastive learning [20, 21].

**Provable guarantees on FPRs even under adversarial attacks without distributional assumptions** By integrating methods from the conformal prediction literature [22, 23] into our RAW framework, we showcase its ability to offer rigorous, *distribution-free* assurances regarding the FPRs. Additionally, we develop a novel technique, inspired by the randomized smoothing [24, 25], to further enhance our provable guarantees. This technique ensures *certified* guarantees on FPRs under arbitrary perturbations, including those adversarial with bounded norms. That is, as long as any transformations on test images stay within a predefined range, our guarantee on FPRs remains valid.

**Extensive empirical studies on various datasets & extension to video watermark** We evaluate the effectiveness of our proposed method across various generative data scenarios, such as the DBDiffusion [26] and MS-COCO [27]. Additionally, we discuss how to extend the proposed framework for video watermarking and provide empirical results to demonstrate its effectiveness. Our assessment includes detection performance, robustness against image manipulations/attacks, the computational efficiency of watermark injection, and the quality of generated images. Experimental results consistently affirm the excellent performance of our approach, as evidenced by significantly improved watermark encoding speed and notable enhancements in detection performance against state-of-the-art adversarial attacks aimed at removing watermarks.

## 1.2 Related Work

**Classical watermarking techniques for images** Image watermarking has long been a fundamental problem in both signal processing and computer vision literature [9, 28]. Methods for image-based watermarking typically operate within either the spatial or frequency domains [9, 29]. Within the spatial domain, methodologies span from basic approaches, such as the manipulation of the least significant bit of individual pixels, to more complex strategies like Spread Spectrum Modulation [30, 31]. In the frequency domain, watermark embedding [9, 32] involves modifying coefficients generated by transformations such as the Discrete Cosine Transform [33].

**Image watermarking using deep learning & Watermark for diffusion models** The advent of advanced deep learning techniques has opened up new avenues for watermarking. Many of these methods [34, 35, 36, 10, 17, 11], are based on the encoder-decoder architecture. In this model, the encoder embeds a binary code into images in latent representations, while the decoder takes an image as input and generates a binary code for comparison with the binary code injected for watermark verification. For example, the HiDDeN technique [36] involves the simultaneous training of encoder and decoder networks, incorporating noise layers specifically crafted to simulate image perturbations. While these methods improve robustness over traditional watermarking, they might not be suitable for real-time watermark injection due to the lengthy feed-forward process in the encoder, especially with larger architectures. Most recently, there has been a line of work focusing on designing watermarks specifically for diffusion models [5, 8, 37]. These methods benefit from improved robustness due to their model-specific design, but the applicability to other types of models is not yet well established.

**Watermarks for protecting model intellectual property** Deep neural networks represent valuable intellectual assets, given the significant resources invested in their training and data collection processes [1]. For example, training stable diffusion models consumes approximately 150,000 GPU hours, amounting to roughly \$600,000 in costs [38]. Given their broad applications in real-world scenarios, ensuring copyright protection and facilitating their identification is crucial in both normal and adversarial contexts [39, 40]. Some approaches embed watermarks directly into model parameters [41], necessitating white-box access for inspection. Others [42, 43] rely on backdoor attacks [44, 45, 46, 47], injecting triggers into training data to enable targeted predictions during testing. These methods primarily focus on safeguarding model intellectual property rather than the generated outputs.

## 2 Preliminary

**Notations.** We consider the problem of embedding watermarks into images and then detecting the watermarks as a binary classification problem. Let $\mathcal{X} = [0, 1]^{C \times W \times H}$ be the input space, with $C$, $W$, and $H$ being the channel, width and height of images, respectively. We denote $\mathcal{Y} = \{0, 1\}$ to be the label space, with label 0 indicating unwatermarked and 1 indicating watermarked versions, respectively. For a vector $v$, we use $\|v\|$ to denote its $\ell_2$-norm.

**Threat Model.** We consider the following use scenario of watermarks between a third-party user Alice, e.g., an artist, a generative model provider Bob, e.g., DALLE-2 from OPENAI, and an adversary Cathy. **(1)** Alice selects a diffusion model (DM) from Bob's API interface and sends an input (e.g., a prompt for text-to-image diffusion models) to Bob for generating images. **(2)** Upon receiving Bob generated images $X \in \mathcal{X}$, Alice embeds a watermark into $X$, denoted as $X' \in \mathcal{X}$, and releases it to the public. **(3)** Adversary Cathy applies (adversarial) image transformation(s), e.g., rotating and cropping, on $X'$ to obtain a modified version $\widetilde{X}'(\in \mathcal{X})$. **(4)** Alice decides if $\widetilde{X}'$ was generated by herself or not.

**Problem Formulation.** From the above, the watermark problem for Alice essentially boils down to a binary classification or hypothesis testing problem:

$$\mathrm{H}_0 : \widetilde{X}' \text{ was generated by Alice (Watermarked) ;}$$
$$\mathrm{H}_1 : \widetilde{X}' \text{ was NOT generated by Alice (Unwatermarked) .}$$

To address this problem, Alice will build a detector given by

$$g(\mathcal{V}_\theta(X)) = \begin{cases} 1(\text{Watermarked}) & \text{if } \mathcal{V}_\theta(X) \geq \tau, \\ 0(\text{Unwatermarked}) & \text{if } \mathcal{V}_\theta(X) < \tau, \end{cases} \tag{1}$$

where $\tau \in \mathbb{R}$ is a threshold value and $\mathcal{V}_\theta$ (to be defined later) is a scoring function which takes the image input and output a value in $[0, 1]$ to indicate its chance of being a sample generated by Alice.

**Alice's Goals.** Alice's objective is to design watermarking algorithms that fulfill the following objectives: **(1) Quality:** the quality of watermarked images should closely match that of the original, unwatermarked images; **(2) Identifiability:** both watermarked and unwatermarked content should be accurately distinguishable; **(3) Robustness:** the watermark should be resilient against various image manipulations.

**Cathy's (Adversary) Goals.** Cathy aims to design attack algorithms to meet the following objectives: **(1) Watermark Removal:** the watermarks embedded by Alice can be effectively eliminated after the attacks; **(2) Quality:** the attacks cannot significantly alter the images.

## 3 RAW

In this section, we formally introduce our RAW framework. At a high level, the RAW framework comprises two consecutive stages: a training stage and an inference stage, as outlined in Figure 2 in Appendix. In the following subsection, we first provide an in-depth description of the training stage.

### 3.1 Training stage

Suppose Alice obtains a batch of diffusion model-generated images. The unwatermarked data are denoted as $\mathcal{D}^{\mathrm{uwm}} \triangleq \{X_i\}_{i=1}^n$ for $i = 1, \ldots, n$. Alice will need to embed watermarks into these images to protect intellectual property.

**Definition 3.1** (Watermarking Module). A watermarking module $\mathcal{E}_{\boldsymbol{w}}(\cdot)$ maps $\mathcal{X}$ to itself, with parameters $\boldsymbol{w} \in \mathbb{R}^{d_1}$.

The watermarking module can take the form of an encoder with an attention mechanism, as seen in the RivaGan [10], or it can involve Fast Fourier Transformation (FFT) followed by frequency adjustments and an inverse FFT, as employed in DwtDct.

In our RAW framework, we propose to add two distinct watermarks into both frequency and spatial domains:

$$\mathcal{E}_{\boldsymbol{w}}(X) = \mathcal{F}^{-1}(\mathcal{F}(X) + c_1 \times v) + c_2 \times u, \tag{2}$$

where $v, u \in \mathcal{X}$ are two watermarks, $c_1, c_2 > 0$ determine the visibility of these watermarks, and $\mathcal{F}(\mathcal{F}^{-1})$ represents the Fast Fourier Transformation (FFT) (inverse FFT), respectively. For simplicity of notation, in the rest of this paper, we will denote $\boldsymbol{w} \triangleq [u, v]$. For implementation, we clip the watermarked image $\mathcal{E}_{\boldsymbol{w}}(X)$ to be within the range $[0, 1]$, making $\mathcal{E}_{\boldsymbol{w}}(X)$ a non-linear function of $X$.

The rationale for adopting the above approach is to simultaneously enjoy the distinct advantages offered by watermarks in both domains. In particular, the incorporation of watermark patterns

in the frequency domain has been widely recognized for its effectiveness against certain image manipulations such as translations and resizing. Moreover, our empirical validation corroborates the improved robustness of spatial domain watermarking in the presence of noise perturbations.

We denote the watermarked dataset $\mathcal{E}_{\boldsymbol{w}}$ to be $\mathcal{D}^{\mathrm{wm}} \triangleq \{\mathcal{E}_{\boldsymbol{w}}(X_i)\}_{i=1}^n$ for $i = 1, \ldots, n$. Alice now wishes to distinguish the combined dataset $\mathcal{D} \triangleq \mathcal{D}^{\mathrm{uwm}} \bigcup \mathcal{D}^{\mathrm{wm}}$ with a verification module, which is a binary classifier.

**Definition 3.2** (Verification Module). A verification module is a mapping $\mathcal{V}_\theta(\cdot) : \mathcal{X} \mapsto [0, 1]$ parameterized by $\theta \in \mathbb{R}^{d_2}$.

The score generated by the verification module for an input image can be understood as the chance of this image being generated by Alice.

To fulfill Alice's first two goals, Alice will consider *jointly* training the watermarking and verification modules parameterized by $\boldsymbol{w}$ and $\theta$, with the following loss function:

$$\mathrm{BCE}(\mathcal{D}) \triangleq \sum_{X \in \mathcal{D}} Y \log(\mathcal{V}_\theta(X)) + (1 - Y) \log(1 - \mathcal{V}_\theta(X)), \tag{3}$$

where $X$ is the training image and $Y \in \{0, 1\}$ is the label indicting $X$ is watermarked or not.

Recall that Alice also aims to enhance the robustness of the watermark algorithms. As a result, we consider transforming the combined datasets with different data augmentations $\mathcal{M}_1, \ldots, \mathcal{M}_m$ to obtain $\mathcal{D}^1 \triangleq \mathcal{M}_1(\mathcal{D}), \ldots, \mathcal{D}^m \triangleq \mathcal{M}_m(\mathcal{D})$, respectively. Here, the data augmentations $\mathcal{M}_1, \ldots, \mathcal{M}_m$ are defined as follow.

**Definition 3.3** (Modification Module). An image modification module is a map $\mathcal{M} : \mathcal{X} \mapsto \mathcal{X}$.

To sum up, the overall loss function for our RAW framework is specified as:

$$\mathcal{L}_{\mathrm{raw}} \triangleq \mathrm{BCE}(\mathcal{D}) + \sum_{k=1}^m \mathrm{BCE}(\mathcal{D}^k), \tag{4}$$

where $\mathrm{BCE}(\cdot)$ denotes the binary cross entropy loss as specified in Equation (3). The loss function above is composed of two terms: $\mathrm{BCE}(\mathcal{D})$, which corresponds to the cross-entropy on the original combined datasets $\mathcal{D}$, and $\sum_{k=1}^m \mathrm{BCE}(\mathcal{D}^k)$, signifying the cross-entropy on the augmented datasets $\mathcal{D}^1, \ldots, \mathcal{D}^m$. In our experiments, inspired by contrastive learning such as those presented in [20, 21], we adopt a two-view data augmentation approach by setting $m = 2$.

### 3.1.1 Overall Training Algorithm

We describe the overall training algorithm below. For completeness, we provide the full pseudo-code as summarized in Algorithm 3 in the appendix. We consider conducting the following two steps alternatively.

- Optimize the verification module $\mathcal{V}_\theta$ based on the overall loss $\mathcal{L}_{\mathrm{raw}}$ by stochastic gradient descent (SGD):
$$\theta_{t+1} \leftarrow \theta_t - \mu_t \nabla_\theta \mathcal{L}_{\mathrm{raw}}(\theta_t, \boldsymbol{w}),$$
where $\mu_t > 0$ is the step size at each step $t$.

- Optimize the watermark $\boldsymbol{w}$ based on $\mathcal{L}_0$, defined as the first term on the right-hand side of Equation (4), with sign-based stochastic gradient descent (SignSGD):
$$\boldsymbol{w}_{t+1} \leftarrow \boldsymbol{w}_t - \nu_t \operatorname{sign}\left(\nabla_{\boldsymbol{w}} \mathcal{L}_0(\theta, \boldsymbol{w}_t)\right), \tag{5}$$
where $\operatorname{sign}(\cdot)$ outputs the sign of each component, and $\nu_t > 0$ is the step size.

We provide two remarks on the optimization procedure for the watermark update, as outlined in Eq. (5). First, we choose signSGD over conventional SGD, based on empirical findings suggesting that (sign-based) first-order methods can improve both training and test performance in data-level optimization tasks [19, 48, 49]. In our context, this refers to cases where the optimized variable is the watermark rather than the model parameters. Second, we focus the watermark optimization solely on the first term because the other terms, $\mathrm{BCE}(\mathcal{D}^k)$ for $k = 1, 2, \ldots, m$, may contain non-differentiable elements with respect to the watermark. These non-differentiable parts make gradient-based optimization impractical.

## 3.2 Inference Stage

We present a generic approach for Alice to obtain provable guarantees on the FPRs when using the previously trained $\mathcal{V}_\theta$ on test images, even amidst adversarial perturbations.

To begin with, we first examine a scenario where the future test data $X_{\text{test}} \in \mathcal{X}$ adheres to an IID pattern with the watermarked data $\mathcal{D}^{\text{wm}}$ generated by Alice, without undergoing any image modifications. In this case, Alice can employ conformal prediction to establish provable guarantees on the FPRs. The main idea is that, by utilizing the trained $\mathcal{V}_\theta$ as a scoring mechanism, the empirical quantile of the watermarked data's distribution will converge to the population counterpart. This convergence is guaranteed by the uniform convergence of cumulative distribution functions (CDFs). To be more specific, we set the threshold $\tau$ defined in Equation (1) to be the $\alpha$-quantile (with finite-sample corrections) of the predicted scores for watermarked data, $\mathcal{V}_\theta(\mathcal{D}^{\text{wm}}) \triangleq \{\mathcal{V}_\theta(\mathcal{E}_{\boldsymbol{w}}(X_i))\}_{i=1}^n$. Following [23, 46], it can be shown that, with high probability, the probability of the resulting detector $g$ misclassifying a watermarked image $X_{\text{test}}$ is upper-bounded by $\alpha$, given that $X_{\text{test}}$ is IID with $\mathcal{D}^{\text{wm}}$. For completeness, a rigorous statement and its proof are provided in Appendix A.2.

The above argument assumes that the future test image $X_{\text{test}}$ follows an IID pattern with the original watermarked data $\mathcal{D}^{\text{wm}}$. However, if the test image $X_{\text{test}}$ undergoes manipulation or attack, denoted by $\mathcal{A}(X_{\text{test}})$, with $\mathcal{A} : \mathcal{X} \mapsto \mathcal{X}$ being an adversarial image manipulation, then it can deviate from the distribution of $\mathcal{D}^{\text{wm}}$. This deviation from IID will render the previous argument invalid. Moreover, in practice, Alice is unaware of the adversarial transformation $\mathcal{A}$ employed by the attacker, thus making it even more challenging to control the FPRs.

To address this problem, we propose to consider a robust version of the originally trained $\mathcal{V}_\theta$, denoted as $\tilde{\mathcal{V}}$, such that $X$ and $\mathcal{A}(X)$ stay close under $\tilde{\mathcal{V}}$, namely

$$|\tilde{\mathcal{V}}(X) - \tilde{\mathcal{V}}(\mathcal{A}(X))| \leq \eta, \tag{6}$$

for all $X$ and a small $\eta > 0$. The reason for finding such $\tilde{\mathcal{V}}$ is because we can relate $\tilde{\mathcal{V}}(\mathcal{A}(X_{\text{test}}))$ back to $\tilde{\mathcal{V}}(X_{\text{test}})$ which is IID with $\tilde{\mathcal{V}}(\mathcal{D}^{\text{wm}})$ (accessible to Alice) to establish the FPRs with previous arguments.

To develop the robust version from the base $\mathcal{V}_\theta$, we will build upon the following result from the randomized smoothing literature. Denote $\mathcal{N}(\mu, \Sigma)$ to be the normal distribution with mean $\mu$ and covariance $\Sigma$ respectively, and $\Phi^{-1}(\cdot)$ to be the inverse of the cumulative distribution function of a standard normal distribution.

**Lemma 3.4** ([50, 51]). *Let $h : \mathbb{R} \to [0, 1]$ be a continuous function. Let $\sigma > 0$, and $H(X) = \mathbb{E}_{Z \sim \mathcal{N}(0, \sigma^2 I)}[h(X + Z)]$. Then the function $\Phi^{-1}(H(X))$ is $\sigma^{-1}$-Lipschitz with respect to $\ell_2$ norm.*

The above result suggests that for *any* (continuous) base verification module (classifier) $\mathcal{V}_\theta$, we can obtain a smoothed version with

$$\tilde{\mathcal{V}}(X) = \Phi^{-1}\left( \mathbb{E}_{Z \sim \mathcal{N}(0, \sigma^2 I)}[\mathcal{V}_\theta(X + Z)] \right), \tag{7}$$

and it is guaranteed that $|\tilde{\mathcal{V}}(X) - \tilde{\mathcal{V}}(Y)| \leq \sigma^{-1}\|X - Y\|$, for any $X, Y \in \mathcal{X}$. Suppose the attacker employs an adversarial attack $\mathcal{A}$ such that $\|X - \mathcal{A}(X)\| \leq \gamma$. We have

$$|\tilde{\mathcal{V}}(X) - \tilde{\mathcal{V}}(\mathcal{A}(X))| \leq \frac{\gamma}{\sigma}. \tag{8}$$

*Remark* 3.5 ($\mathcal{A}$ can not be excessively adversarial). We emphasize that the transformation $\mathcal{A}$ should not be excessively adversarial. In other words, the parameter $\gamma$ should be a very low value for both theoretical and practical reasons. From a theoretical perspective, an overly adversarial transformation $\mathcal{A}$ can result in trivial TPRs/FPRs. For instance, if watermarked images are transformed into a completely uniform all-white or all-black state, it becomes impossible to detect the watermark. From a practical standpoint, an excessively adversarial transformation $\mathcal{A}$ tends to overwrite the original content within the images. This directly contradicts the intentions of attackers and may not achieve the desired stealthy modifications.

### 3.2.1 Overall Inference Algorithm

Given a pair of $(\mathcal{E}_{\boldsymbol{w}}, \mathcal{V}_\theta)$, a desired robust range $\gamma > 0$, and a smoothing parameter $\sigma > 0$, Alice now will set the thresholding value $\tau$, as introduced in Equation (1), to satisfy:

$$\hat{F}\big(\tau + \frac{\gamma}{\sigma}\big) = \alpha - \sqrt{(\log(2/\delta)/(2n))}, \tag{9}$$

where $\delta \in (0, 1)$ is a violation rate describing the probability that the FPRs exceeds $\alpha$, and $\hat{F}$ is the empirical cumulative distribution function of $\{\tilde{\mathcal{V}}(\mathcal{E}_{\boldsymbol{w}}(X_i))\}_{i=1}^n$, where

$$\tilde{\mathcal{V}}(\mathcal{E}_{\boldsymbol{w}}(X_i)) \triangleq \Phi^{-1}\bigg(\mathop{\mathbb{E}}_{Z \sim \mathcal{N}(0,\sigma^2 I)}[\mathcal{V}_\theta(\mathcal{E}_{\boldsymbol{w}}(X_i) + Z)]\bigg).$$

The next result shows that if a future test input comes from the same distribution as the watermarked data $\mathcal{D}^{\mathrm{wm}}$, the above procedure can be configured to achieve any pre-specified false positive rate $\alpha$ with high probability.

**Theorem 3.6** (Certified FPRs of $g$ based on threshold in Equation (9)). *Given any watermarked dataset $\mathcal{D}^{wm}$ and its associated verification module $\mathcal{V}_\theta$, suppose that the test data $X_{test}$ are IID drawn from the distribution of $\mathcal{D}^{wm}$. Given any $\delta \in (0, 1)$, $\gamma > 0$ and $\sigma > 0$, for any (adversarial) image transformations $\mathcal{A}$ such that $\|\mathcal{A}(X) - X\| \leq \gamma$ for all $X \in \mathcal{X}$, the robust verification module $\tilde{V}$ specified in Eq. (7) and its corresponding hard-label detector $g(\cdot)$ introduced in Eq. (1), with the threshold $\tau$ as specified in Eq. (9) satisfy*

$$\mathbb{P}\bigg(g(\tilde{V}(\mathcal{A}(X_{test}))) = 0 \, (Unwatermarked) \mid \mathcal{D}^{wm}\bigg) \leq \alpha,$$

*with probability at lease $1 - \delta$ for any $\alpha \in (0, 1)$ such that $\alpha > \sqrt{(\log(2/\delta)/(2n))}$.*

Due to space constraints, the proof is provided in Appendix A.1. The above result shows that by using the decision rule as specified in Equation (9), Alice can obtain a provable guarantee on the Type I error rate in terms of detecting future test input $X_{\mathrm{test}}$ even $X_{\mathrm{test}}$ is adversarially perturbed within $\gamma$-range (as measured by $\ell_2$-norm), under the condition that the future test input $X_{\mathrm{test}}$ is independently and identically distributed as the $\mathcal{D}^{\mathrm{wm}}$, namely watermarked samples generated by the artist. Moreover, the result above only addresses the detector's performance in terms of FPRs and does not account for True Positive Rates (TPRs). To achieve high TPRs, careful selection of $\gamma$ and $\sigma$ is essential. For instance, an excessively large $\sigma$ will cause the smoothed classifier $\tilde{\mathcal{V}}$ to lose its detection capability.

## 4 Experiments

In this section, we conduct a comprehensive evaluation of our proposed RAW, including (1) detection performance, (2) robustness, (3) watermarking speed, (4) the quality of watermarked images, and (5) the provable FPRs guarantees under adversarial attacks. Our findings reveal significantly enhanced robustness in RAW while preserving the quality of generated images. Furthermore, a substantial reduction in watermark injection time, indicates the suitability of RAW for on-the-fly deployment. All the experiments were conducted on machines equipped with Nvidia Tesla A100s.

### 4.1 Experimental setups

**Datasets (1)** In line with the previous work [8], we employ Stable Diffusion-v2-1 [1], an open-source text-to-image diffusion model, with DDIM sampler, to generate images. All the prompts used for image generation are sourced from the MS-COCO dataset [27]. **(2)** We further evaluate our RAW utilizing DBdiffusion [26], a dataset consisting of 14 million images generated by Stable Diffusion. This dataset encompasses a wide array of images produced under various prompts, samplers, and user-defined hyperparameters. Ablation studies examining various generative models, such as SDXL-1.0, are detailed in Appendix D.

**Verification Modules/Classifiers** In terms of verification modules, for all the results reported in the main text, we utilize ResNet 18 [52]. For training the verification modules, 500 images are randomly selected for training for each dataset. Subsequently, we evaluate the trained watermarks and associated models on 1000 new, unwatermarked images and their watermarked versions. Ablation studies on using different models, such as VGG [53] as verification modules, and different number of training data are provided in Section 4.4 and Appendix D.

Table 2: Summary of main results. 'N-ROC' denotes the AUROC performance without image manipulations or adversarial attacks. 'Ad-ROC' represents the average performance across nine distinct image manipulations and attacks. The 'Encoding Speed' column denotes the efficiency of injecting watermarks into images post-training, measured in seconds (CPU time) per image.

| Dataset | Method | Encoding Speed ↓ | N-ROC ↑ | Ad-ROC ↑ | FID ↓ | CLIP ↑ |
|---|---|---|---|---|---|---|
| MS-Coco FID: 24.12 CLIP: 0.382 | DwTDcT | 0.048 | 0.83 | 0.54 | 25.10 | 0.359 |
| | DwTDcTSvd | 0.122 | 0.98 | 0.75 | 25.21 | 0.361 |
| | RivaGan | 1.16 | 0.99 | 0.81 | 24.87 | 0.359 |
| | StegaStamp | 1.45 | 0.99 | 0.93 | 42.31 | 0.291 |
| | RAW (Ours) | **0.0051** | 0.98 | **0.92** | 24.75 | 0.360 |
| DBdiffusion | DwTDcT | 0.048 | 0.81 | 0.55 | 3.12 | 0.427 |
| | DwTDcTSvd | 0.110 | 0.99 | 0.78 | 10.24 | 0.421 |
| | RivaGan | 1.87 | 0.99 | 0.82 | 19.75 | 0.424 |
| | StegaStamp | 1.90 | 0.99 | 0.92 | 40.59 | 0.386 |
| | RAW (Ours) | **0.0078** | 0.98 | **0.90** | 23.48 | 0.425 |

Table 3: AUROC performance of state-of-the-art methods under 9 (adversarial) image manipulations: Rotation $90°$, Cropping and resizing 70%, Gaussian Blur with a kernel size of $(7, 9)$ and bandwidth of 4, Noise with IID mean Gaussian $\sigma = 0.05$, Jitter with brightness factor 0.6, JPEG compression with quality 50, and 3 attacks (VAE Att1, VAE Att2, Diff Att).

| Datasets | MS-COCO | | | | DBDiffusion | | | |
|---|---|---|---|---|---|---|---|---|
| | DwtDct | DwtDctSvd | RivaGan | **Ours** | DwtDct | DwtDctSvd | RivaGan | **Ours** |
| JPEG 50 | 0.61 | 0.99 | 0.99 | 0.91 | 0.50 | 0.95 | 0.99 | 0.99 |
| Rotation $90°$ | 0.51 | 0.55 | 0.39 | 0.96 | 0.47 | 0.54 | 0.38 | 0.84 |
| Cropping 70% | 0.64 | 0.52 | 0.99 | 0.95 | 0.65 | 0.61 | 0.99 | 0.84 |
| Gaussian Blur | 0.52 | 0.92 | 0.99 | 0.93 | 0.53 | 0.99 | 0.99 | 0.98 |
| Gaussian Noise | 0.47 | 0.76 | 0.99 | 0.90 | 0.84 | 0.99 | 0.99 | 0.98 |
| Jittering | 0.65 | 0.78 | 0.98 | 0.96 | 0.47 | 0.68 | 0.98 | 0.99 |
| VAE Att1 | 0.50 | 0.73 | 0.62 | 0.89 | 0.49 | 0.75 | 0.67 | 0.80 |
| VAE Att2 | 0.48 | 0.78 | 0.67 | 0.90 | 0.49 | 0.73 | 0.63 | 0.81 |
| Diff Att | 0.49 | 0.71 | 0.69 | 0.83 | 0.51 | 0.73 | 0.70 | 0.82 |
| Average | 0.54 | 0.75 | 0.82 | **0.92** | 0.56 | 0.78 | 0.82 | **0.90** |

**Watermark Parameters** In Equation (2) of RAW, two parameters $c_1$ and $c_2$ control the invisibility of the watermark, thereby influencing the quality of watermarked images. In the main text results, we set $c_1 = c_2 = 0.05$ to align with the image quality of watermarked images produced by other state-of-the-art methods (refer to the 'FID' and 'CLIP' columns in Table 2). We conduct an ablation study exploring different values of $c_1$ and $c_2$ in Section 4.4 and Appendix D. For all other watermarking techniques, we implement them using the open-source package employed by the Official Stable Diffusion Model.

**Evaluation Metrics (1)** To assess the detection performance, we adhere to the convention of reporting the area under the curve of the receiver operating characteristic (AUROC) [17, 5]. **(2)** For evaluating the quality of the watermarked images, we adopt both the Frechet Inception Distance (FID) [54] and the CLIP score [55], following the methodology outlined in [8]. Additionally, we also evaluate the quality of watermarked images using metrics such as PSNR and SSIM, and include the results in the appendix. For our watermark scheme, each independent run (including training and testing) corresponds to a unique watermark. All metrics are averaged across 5 independent runs, with a standard deviation of 0.03 for all detection performances (AUROC) and 0.7 for image quality metrics.

## 4.2 Main Results

**Detection performance and image generation quality** To ensure a fair comparison, we primarily evaluate our proposed RAW against other model-agnostic approaches, presenting the summarized results in Table 2, along with visual examples illustrated in Appendix D.4. Additionally, for completeness, we compare our proposed RAW against model-specific methods and provide the results in the

appendix. Our RAW exhibits comparable performance to encoder-decoder-based approaches, e.g., RivaGAN, while concurrently achieving similar FID and CLIP scores, which underscores superior image quality compared to alternatives. We report the PSNR and SSIM of the watermarked images in the Table 6 in the appendix. Our method outperforms StegaStamp in PSNR and SSIM but falls behind DwtDctSvd and RivaGAN, as expected. StegaStamp prioritizes robust detection by injecting significant noise, reducing image quality. In contrast, DwtDctSvd and RivaGAN maintain better image quality at the cost of reduced robustness to image perturbations, a point further detailed in the next paragraph.

**Robust detection performance** We assess the robustness of our proposed RAW against six common data augmentations and three adversarial attacks in this subsection. The data augmentation set comprises: color jitter with a brightness factor of $0.5$, JPEG compression with quality $50$, rotation by $90°$, addition of Gaussian noise with $0$ mean and standard deviation $0.05$, Gaussian blur with a kernel size of $(7, 9)$ and bandwidth $4$, and $70\%$ random cropping and resizing. For adversarial attacks, we select three state-of-the-art methods for removing watermarks, with two VAE-based attacks `Bmshj2018` [14] (VAE Att1) and `Cheng2020` [56] (VAE Att2) from CompressAI [57] with compression factors are set to 3 for both models, and one diffusion-model attack with noise steps of 60 following [58].

The averaged results are in the 'AUROC (Adv)' column of Table 2 and the detailed results are summarized in Table 3. Our approach demonstrates superior performance compared with alternative methods. Specifically, across both datasets, the average AUROC for our RAW increased by $70\%$ and $13\%$ for nine image manipulations/attacks, surpassing frequency- and encoder-decoder-based methods. We note that Stegastamp demonstrates similar averaged robust detection performance compared to ours. However, this comes at the expense of reduced image quality, as evidenced by markedly increased FID scores and/or lowered SSIM/PSNR scores.

**Watermark injection speed** We investigate the time costs needed to embed watermarks into images. We note that the watermark injection process occurs post-training. Therefore, our watermark injections only necessitate one FFT, two additions, and another inverse FFT. In Table 4, we present CPU time (in seconds) elapsed for injecting watermarks into different image quantities. Notably, our method achieves substantial time efficiency

Table 4: CPU time (seconds) elapsed for injecting watermarks into images. **Lower** values are preferred.

| Batch Size $\rightarrow$ | 5 images | 100 images | 500 images |
|---|---|---|---|
| DwtDct | 0.27 | 4.8 | 24.5 |
| DwtDctSvd | 0.64 | 12.2 | 60.1 |
| RivaGAN | 5.52 | 116 | $> 500$ |
| StegaStamp | 7.34 | 134 | $> 500$ |
| RAW (Ours) | 0.35 | 0.51 | 0.76 |

improvements, approximately $\mathbf{30\times}$ ($\mathbf{200\times}$) faster than the frequency-based (encoder-decoder based) method, respectively. This is attributed to streamlined batch operations in our RAW. This highlights the suitability of our approach for on-the-fly deployment.

### 4.3 Certified FPRs

We assess the certified FPRs performance of our proposed RAW by varying the FPRs rate $\alpha$ pre-specified by Alice. We set the adversaril radius $\gamma = 0.001$ and the smoothing parameter $\sigma = 0.05$. We summarize the results of five independent runs in Figure 1(a) and report the mean (with standard error $< 0.002$). The results show that the FPRs of RAW consistently matches the theoretical upper bounds (i.e., $\alpha$), supporting the result presented in Theorem 3.6.

### 4.4 Ablation Studies

**Effect of training sample size on prediction performance** We manipulate the sample size of the watermarked training dataset $\mathcal{D}^{\text{wm}}$ to assess its impact on detection performance. These findings are illustrated in Figure 1(b), where we note that satisfactory detection performance can be achieved with a reasonably small training dataset, e.g., 100 images.

**Trade-off between robustness and image quality** We explore the trade-off between robustness and image quality by adjusting the watermark strength parameters $c_1$ ($= c_2$), as illustrated in Figure 1 (c) & (d) below. We note that with increasing values of $c_1$ and $c_2$, the average AUROC, under 9 (adversarial) images manipulations, also increases, while the image quality only exhibits a slight

degradation, as indicated by the slightly increased FID value. These findings also highlight the stability of the watermark hyperparameters $c_1$ and $c_2$ in our proposed RAW, a desirable characteristic for real-world deployments, as preferred by practitioners.

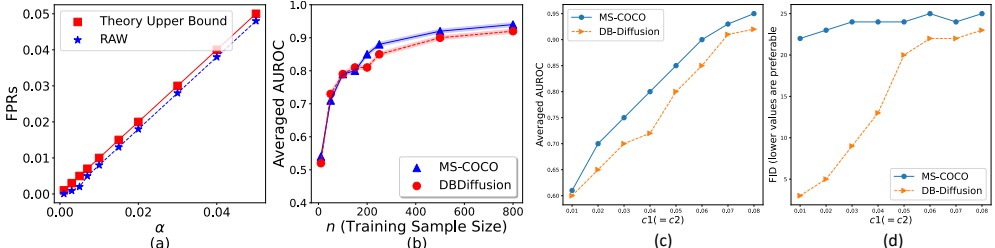

Figure 1: (a) FPRs of our proposed RAW. (b) Impact of training sample size on detection performance. (c) & (d) The tradeoff between the quality of watermarked images, assessed using FID (lower values are preferable), and the detection robustness.

### 4.5 Extension to Video Watermark

We discuss how to extend the proposed watermark framework from AI generated images to videos. The main idea is to embed the same pre-trained watermark either in every frame or at intervals, such as every three frames, of the video. The corresponding verification module, i.e., the watermark classifier, is then used to detect the watermark in each frame where it was embedded. The overall presence of the watermark in the video is determined by the detection results from these frames.

We evaluate the pre-trained (on the MS-COCO dataset) watermark and its corresponding classifier using short videos created by the `stable-video-diffusion-img2vid-xt` model, which generates videos from images. The images used to produce the videos are sourced from the DiffusionDB dataset. This guarantees that the testing videos were not exposed to either the watermark or the classifier during the training process. The results are summarized in Table 5 below, where we observed that our proposed method achieves a similar detection rate compared to RivaGan, while providing a significantly faster watermark encoding speed ($\times 60$ times faster).

Table 5: Encoding Speed (CPU Only) and AUROC (over fresh 500 test samples) for video watermark with the proposed RAW

| Method | Video Resolution | Number of Frames | Time Elapsed | AUROC |
| --- | --- | --- | --- | --- |
| **RAW (Ours)** | $512 \times 512$ | 24 | **0.2 - 0.5s** | 0.96 |
| RivaGan | $512 \times 512$ | 24 | 8 - 12s | 0.97 |

## 5 Conclusion

In this study, we introduce the RAW framework as a versatile watermarking approach essential for protecting intellectual property and mitigating potential misuse of AI-generated images. The proposed RAW framework offers several notable features, including significantly enhanced watermark encoding speed and/or detection performance, along with the assurance of provable guarantees on false positive rates even under adversarial perturbations in test images. Experimental findings across various datasets validate its advantages.

**Limitation & Future Work** One potential limitation of the proposed method is that when applied to large-scale systems with millions of users, the associated training loss will be high. One interesting direction is to study how to efficiently fine-tune from one pair of watermark and its corresponding verification module to a new pair. Other directions include investigating the maximum number of concurrent watermarks learnable in a single training session and optimal smoothing strategies for wider certified radii.

**Broader Impact** This paper aims to contribute to the advancement of trustworthy machine learning, particularly in ensuring the safe and legitimate use of contemporary generative artificial intelligence. Our efforts could have several positive societal implications, such as protecting intellectual property and preventing potential misuse of AI-generated images.

The **Appendix** contains proofs, experimental details, and ablation studies.

## Acknowledgment

The work of Xun Xian, Xuan Bi, and Mingyi Hong was supported in part by a sponsored research award by Cisco Research. The work of Ganghua Wang and Jie Ding was supported in part by the Army Research Office Early Career Program Award under grant number W911NF2310315.

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

# Appendix for RAW: A Robust and Agile Plug-and-Play Watermark Framework for AI-Generated Images with Provable Guarantees

In Section A, we outline the formal proofs supporting the theoretical findings introduced in the main text. In Section B, we present omitted details, encompassing pseudocode and additional deliberations regarding the construction of RAW. In Section C, we offer implementation details of our experiments. In Section D, we present additional ablation studies on different hyperparameters.

## A  Proof of theoretical results

In this section, we provide the proof for Theorem 3.6. To begin with, we first revisit the overall inference procedure employed by our RAW and provide several detailed discussions. For the reader's convenience, we include the pseudo-code outlining the inference stage protocol of RAW in Algorithm 1 below.

---

**Algorithm 1** Conformal (Inference) Watermark Detection

---

**Input:** querying input $X_{\text{test}}$, watermarked dataset $D^{\text{wm}} = \{\mathcal{E}_{\boldsymbol{w}}(X_i)\}_{i=1}^n$, verification module $\mathcal{V}_\theta$, desired false positive rate $\alpha \in (0, 1)$, violation rate $\delta \in (0, 1)$, desired adversarial robust range $\gamma > 0$, smoothing parameter $\sigma > 0$, and the smoothed/robust verification module $\tilde{\mathcal{V}}(\cdot) \triangleq \Phi^{-1}(\mathbb{E}_{Z \sim \mathcal{N}(0, \sigma^2 I)}[\mathcal{V}_\theta(\mathcal{E}_{\boldsymbol{w}}(\cdot) + Z)])$

---

1: Receiving a future query sample $X_{\text{test}}$
2: **for** $i = 1$ to $n$ **do**
3:     Calculate $s_i \triangleq \tilde{\mathcal{V}}(\mathcal{E}_{\boldsymbol{w}}(X_i))$ // $X_i \in D^{\text{wm}}$
4: **end for**
5: Select the decision threshold $\tau$ according to Equation (10).
6: Determine if $X_{\text{test}}$ is watermarked if $\tilde{\mathcal{V}}(\mathcal{E}_{\boldsymbol{w}}(X_{\text{test}})) \geq \tau$

---

**Output:** The decision if the sample $X_{\text{test}}$ is a watermarked sample or not

---

The overall inference process for the proposed RAW mainly involves determining a decision threshold value $\tau$. This threshold is calculated based on the provided watermarked dataset $D^{\text{wm}}$ and the corresponding (smoothed) verification module $\tilde{\mathcal{V}}$, and it should satisfy the following condition:

$$\hat{F}(\tau + \frac{\gamma}{\sigma}) = \alpha - \sqrt{\frac{\ln \frac{2}{\delta}}{2n}}, \tag{10}$$

where $\delta \in (0, 1)$ is the violation rate describing the probability that the (FPRs) exceeds $\alpha$, and $\hat{F}$ is the empirical cumulative distribution function of the watermarked dataset under $\tilde{\mathcal{V}}$, i.e., $\{\tilde{\mathcal{V}}(\mathcal{E}_{\boldsymbol{w}}(X_i))\}_{i=1}^n$. In the case where $\sqrt{(\log(2/\delta)/(2n))} > \alpha$, we set the thresholding value $\tau$ to be the maximum of $\tilde{\mathcal{V}}(\mathcal{E}_{\boldsymbol{w}}(X_i))\}_{i=1}^n$. We note that under such cases there will be no theoretical guarantees in term of FPRs.

### A.1  Proof of Main Results

The proof of Theorem 3.6 is built upon the techniques established in our previous work [46]. In [46], we developed similar provable guarantees under the IID assumption of future test data. However, in our case of Theorem 3.6, such an IID assumption is no longer valid and hence raise up new technical challenges. To prove Theorem 3.6, we will use the following result concerning the convergence properties of empirical cumulative distribution functions (ECDFs).

**Lemma A.1** (Dvoretzky–Kiefer–Wolfowitz inequality). *Given a natural number $n$, let $X_1, X_2, \ldots, X_n$ be real-valued independent and identically distributed random variables with cumulative distribution function $F(\cdot)$. Let $\hat{F}(\cdot)$ denote the associated empirical distribution function.*

*The interval that contains the true CDF, $F(x)$, with probability $1 - \delta$ is specified as*

$$\hat{F}(x) - \varepsilon \leq F(x) \leq \hat{F}(x) + \varepsilon \ where \ \varepsilon = \sqrt{\frac{\ln \frac{2}{\delta}}{2n}}.$$

*Proof of Theorem 3.6.* Note that $X_{\text{test}}$ is IID drawn from the watermarked data distribution and we have

$$\mathbb{P}(g(\mathcal{A}(X_{\text{test}})) = 0 \ (\text{Unwatermarked}) \mid \mathcal{D}^{\text{wm}})$$

$$= \mathbb{P}(\tilde{\mathcal{V}}(\mathcal{A}(X_{\text{test}})) \leq \tau \mid \mathcal{D}^{\text{wm}}) \tag{11}$$

$$\leq \mathbb{P}(\tilde{\mathcal{V}}(X_{\text{test}}) - \frac{\gamma}{\sigma} \leq \tau \mid \mathcal{D}^{\text{wm}}) \tag{12}$$

$$= \mathbb{E}_{X_{\text{test}}} \mathbf{1}\{\tilde{\mathcal{V}}(X_{\text{test}}) \leq \tau + \frac{\gamma}{\sigma} \mid \mathcal{D}^{\text{wm}}\}$$

$$= \mathbb{E}_{X_{\text{test}}} \mathbf{1}\{F(\tilde{\mathcal{V}}(X_{\text{test}})) \leq F(\tau + \frac{\gamma}{\sigma}) \mid \mathcal{D}^{\text{wm}}\} \tag{13}$$

$$= \mathbb{P}(F(\tilde{\mathcal{V}}(X_{\text{test}})) \leq F(\tau + \frac{\gamma}{\sigma}) \mid \mathcal{D}^{\text{wm}})$$

$$\leq \mathbb{P}(F(\tilde{\mathcal{V}}(X_{\text{test}})) \leq \hat{F}(\tau + \frac{\gamma}{\sigma}) + \varepsilon \mid \mathcal{D}^{\text{wm}}) \quad (\varepsilon = \sqrt{\frac{\ln \frac{2}{\delta}}{2n}}) \tag{14}$$

$$= \alpha - \varepsilon + \varepsilon \tag{15}$$

$$= \alpha,$$

holds with probability at least $1 - \delta$. The equation (11) is because of the decision rule as specified in Algorithm 1, and the inequality (12) is due to the lipschitz condition of $\tilde{\mathcal{V}}$ with parameter $\sigma^{-1}$, namely

$$|\tilde{\mathcal{V}}(X) - \tilde{\mathcal{V}}(Y)| \leq \sigma^{-1} \|X - Y\|,$$

for any $X, Y \in \mathcal{X}$. Additionally, the $F$ in equation (13) represents the CDF of the watermarked data under $\tilde{\mathcal{V}}(\cdot)$, i.e., $\tilde{\mathcal{V}}(\mathcal{E}_{\boldsymbol{w}}(X))$, while $\hat{F}$ in (14) denotes the empirical CDF obtained from $\mathcal{D}^{\text{wm}}$, i.e., $\{\tilde{\mathcal{V}}(\mathcal{E}_{\boldsymbol{w}}(X_i))\}_{i=1}^n$ under $\tilde{\mathcal{V}}(\cdot)$. The inequality in (14) arises from the DKW inequality as specified in Lemma A.1. Furthermore, the Equation (15) is based on the fact that the CDF follows a uniform distribution (a result of the probability integral transformation) and the selection of the thresholding value specified in Equation (10). □

## A.2  Provable FPRs without adversarial attacks

In this section, we present the omitted results of the provable FPRs concerning the watermark detection performance of our RAW in the absence of adversarial attacks, as elaborated in Line 225 of the main text. All the techniques used in this section are based on our previous work [46], as discussed earlier.

As there are no anticipated adversarial attacks on test images, there is consequently no requirement to apply smoothing/robustification to the trained verification model $\mathcal{V}_\theta$. The corresponding pseudo-code is outlined in Algorithm 2 below. Similarly, the updated thresholding value $\tau$ is selected to satisfy the

---

**Algorithm 2** Conformal (Inference) Watermark Detection under no adversarial attacks

---

**Input:** querying input $X_{\text{test}}$, watermarked dataset $D^{\text{wm}} = \{(\mathcal{E}_{\boldsymbol{w}}(X_i))\}_{i=1}^n$, verification module $\mathcal{V}_\theta$, desired false positive rate $\alpha \in (0, 1)$, violation rate $\delta \in (0, 1)$

---

1: Receiving a future query sample $X_{\text{test}}$
2: **for** $i = 1$ to $n$ **do**
3:    Calculate $s_i \triangleq \mathcal{V}_\theta(\mathcal{E}_{\boldsymbol{w}}(X_i))$ // $X_i \in D^{\text{wm}}$
4: **end for**
5: Select the decision threshold $\tau$ according to Equation (10).
6: Determine if $X_{\text{test}}$ is watermarked if $\mathcal{V}_\theta(\mathcal{E}_{\boldsymbol{w}}(X_{\text{test}})) \geq \tau$

---

**Output:** The decision if the sample $X_{\text{test}}$ is a watermarked sample or not

---

condition.

$$\hat{F}(\tau) = \alpha - \sqrt{\frac{\ln \frac{2}{\delta}}{2n}}, \tag{16}$$

where $\delta \in (0, 1)$ is the violation rate describing the probability that the (FPRs) exceeds $\alpha$, and $\hat{F}$ is the empirical cumulative distribution function of the watermarked dataset under $\mathcal{V}_\theta$, i.e., $\{\mathcal{V}_\theta(\mathcal{E}_{\boldsymbol{w}}(X_i))\}_{i=1}^n$. Contrasting the selection of $\tau$ under adversarial attacks in Equation (10), we note that the term $\gamma/\sigma$ is omitted because of the absence of adversarial attacks. In the case where $\sqrt{(\log(2/\delta)/(2n))} > \alpha$, we set the thresholding value $\tau$ to be the maximum of $\{\mathcal{V}_\theta(\mathcal{E}_{\boldsymbol{w}}(X_i))\}_{i=1}^n$.

**Theorem A.2** (Certified FPRs under no adversarial attacks). *For any watermarked dataset $\mathcal{D}^{wm}$ and its associated verification module $\mathcal{V}_\theta$, suppose that the test data $(X_{test}, Y_{test})$ are IID drawn from the distribution of $\mathcal{D}^{wm}$. Given $\delta \in (0, 1)$, the detector $g(\cdot)$ (defined in Line 231 in the main text) with the threshold $\tau$ as specified in Equation (16) satisfies*

$$\mathbb{P}(g(X_{test})) = 0 \text{ (Unwatermarked)} \mid \mathcal{D}^{wm}) \leq \alpha$$

*with probability at least $1 - \delta$ for any $\alpha \in (0, 1)$ such that $\alpha > \sqrt{(\log(2/\delta)/(2n))}$.*

*Proof.* The proof follows the same approach as the proof for Theorem 1, with the exclusion of the $\gamma/\sigma$ term. $\qquad\square$

# B  Omitted Details and Further Discussions

In this section, we initially present the omitted pseudo-code for the training algorithm, followed by further discussions regarding the design of RAW.

## B.1  Overall flow of the proposed RAW

Figure 2 below demonstrates the overall flow of our RAW framework and its differences from encoder-decoder-based methods. In RAW, watermarks are directly introduced and injected into images and are jointly trained with the watermark classifier.

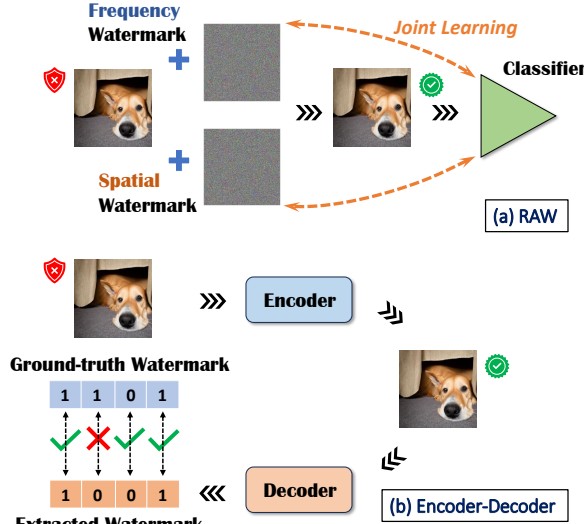

Figure 2: The overall flow of the proposed RAW framework.

## B.2  Pseudocode for training algorithms

The pseudocode for the overall training pipeline of RAW is outlined in Algorithm 3.

---
**Algorithm 3** Training Algorithms for RAW

---

**Input:** (I) Image sets generated from a diffusion model $\{X_i\}_{i=1}^n$; (II) watermark visibility parameter $c_1, c_2$; (III) learning rates $\{\mu_t\}_{t=1}^T, \{\nu_t\}_{t=1}^T$.
   *Initialize*: (1) a verification module $\mathcal{V}_\theta : \mathcal{X} \mapsto [0, 1]$, (2) a watermarking module: $\mathcal{E}_{\boldsymbol{w}}(X) = \mathcal{F}^{-1}(\mathcal{F}(X) + c_1 \times v) + c_2 \times w$ with each entries in $u, v \in \mathcal{X}$ initialized as IID uniform random variables.

---

1: **for** $i = 1$ to $T$ **do**
2:     Clipping the watermarked data to be within the range $[0, 1]$;
3:     Given $\mathcal{V}_\theta$, optimizing $\boldsymbol{w}$ based on $\mathcal{L}_{\text{raw}}$ with SignSGD;
4:     Given the watermark $\boldsymbol{w}$, updating $\theta$ based on $\mathcal{L}_{\text{raw}}$ with SGD;
5: **end for**

---

**Output:** (1) The verification module $\mathcal{V}_\theta$; (2) Watermarking method $\mathcal{E}_{\boldsymbol{w}}$

---

## B.3  Additional metrics on image quality

We report the PSNR and SSIM of the watermarked images in the Table 6 below. For PSNR and SSIM, our method outperforms the StegaStamp method but is behind DwtDctSvd and RivaGAN. These results are consistent with our expectations. The StegaStamp method achieves more robust detection by sacrificing image quality through injecting a large amount of noise into the image, while DwtDctSvd and RivaGAN maintain image quality slightly better at the cost of reduced robustness against (adversarial) image perturbations.

| Dataset | MS-COCO | | DBdiffusion | |
|---|---|---|---|---|
| **Method** | **PSNR** ↑ | **SSIM** ↑ | **PSNR** ↑ | **SSIM** ↑ |
| **RAW (Ours)** | 29.1 | 0.94 | 29.0 | 0.93 |
| StegaStamp | 27.1 | 0.90 | 28.1 | 0.90 |
| DwtDctSvd | 39.6 | 0.98 | 37.8 | 0.97 |
| RivaGAN | 36.4 | 0.96 | 38.5 | 0.97 |

Table 6: Additional metrics for image quality.

## B.4 Further Discussions

We now elaborate on two pivotal aspects of our watermark designs and overarching training algorithms: **(I)** the joint training scheme for watermarking and verification modules, and **(II)** the integration of spatial-domain watermarks.

**(I) The joint training scheme for watermarking and verification modules.** Theoretically, using standard arguments from classical learning theory [59], it can be shown that training both the watermarking and the verification modules to distinguish between watermarked and unwatermarked data will not lead to a test accuracy worse than when the watermark is fixed, and only the model is trained. From a practical perspective, the initially randomly initialized watermarks may not align well with specific training data, emphasizing the need to optimize watermarks for distinct data scenarios. Our empirical observations support this notion, as evidenced in Figure 5 (left), where the joint training scheme leads to a significantly higher test accuracy and lower training loss compared with the scenario where the watermark is fixed.

**(II) The inclusion of spatial domains.** Classical methods for embedding watermarks primarily introduce them into the frequency domains of images [13]. However, it has been empirically observed that such watermarks are susceptible to manipulations, such as Gaussian noise [8]. To overcome this vulnerability, we draw inspiration from the model reprogramming literature [60], where watermarks are incorporated into the spatial domain to enhance accuracy in distinguishing in- and out-distribution data [49]. Consequently, we explore the integration of watermarks into the spatial domain (in addition to the frequency domain), as outlined in Equation (2). We empirically observed that including spatial watermarks could significantly boost the test accuracy of the trained verification module under Gaussian-noise manipulations on test data, as depicted in Figure 5 (right).

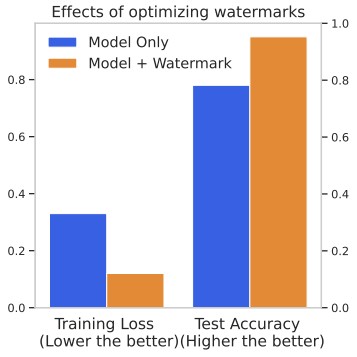 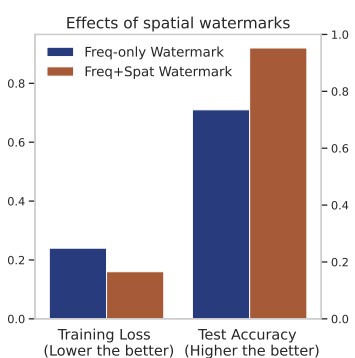

Figure 3: Effects of (1) joint training and (2) spatial watermarks.

## C Implementation details

In this section, we first list the implementation details for the results presented in the main text. Next, we present ablation studies on the performance of our RAW by using different hyper-parameters.

### C.1 Implementation Details

**Watermark setup** Recall that the watermarking module in our RAW takes the form of

$$\mathcal{E}_{\boldsymbol{w}}(X) = \mathcal{F}^{-1}(\mathcal{F}(X) + c_1 \times u) + c_2 \times v,$$

where $u, v \in \mathcal{X}$ are two watermarks injected into spatial and frequency domains, respectively. For the results presented in the main text, we set the watermark strength parameters $c_1$ and $c_2$ to be $0.05$ each. To enhance invisibility further, we implement a circular mask with a radius of $150$ for the frequency domain watermarks, inspired by the procedure outlined in [8].

**Verification module setup** In terms of the verification module $\mathcal{V}_\theta$, throughout all the experiments in the main text, we adopt the pre-trained ResNet 18 [52] architecture.

**Training data augmentation** Two types of data augmentations are employed during the training phase. (1) To enhance the robustness of the trained verification model $\mathcal{V}_\theta$ against Gaussian noise, ensuring its sustained predictive effectiveness post-smoothing, we introduce Gaussian noise into the training data. This augmentation entails the addition of noise with a mean of zero and a standard deviation of $0.5$ during the training process. (2) Additionally, we employ standard image augmentations such as random cropping and flipping to facilitate training.

## D Additional Ablation Studies

### D.1 Verification module/model architectures.

In this section, we assess the watermark detection performance of our RAW using diverse model architectures: ResNet 9, ResNet 34, VGG 16 [53] and ViT [61]. The summarized results on both Stable-Diffusion-generated MS-COCO and DBDiffusion dataset can be found in Table 7 below. FID and CLIP scores are omitted, given that the watermark strength parameters $c_1$ and $c_2$ are consistent with those in the main text (i.e., both set to $0.5$). Notably, we observed an improvement in both benign and adversarial detection performances with a more complex model, ResNet 34, which is reasonable as complex models often possess greater learning capacity.

Table 7: Summary of detection results under different model architectures. AUROC (Ben) denotes the AUROC performance without image manipulations or adversarial attacks. AUROC (Adv) represents the average performance across nine distinct image manipulations and attacks.

| Dataset → | MS-COCO | | DBDiffusion | |
|---|---|---|---|---|
| | AUROC (Ben) ↑ | AUROC (Adv) ↑ | AUROC (Ben) ↑ | AUROC (Adv) ↑ |
| ResNet 9 | 0.94 | 0.82 | 0.90 | 0.80 |
| ResNet 34 | 0.99 | 0.93 | 0.99 | 0.94 |
| VGG 16 | 0.99 | 0.93 | 0.99 | 0.90 |
| ViT | 0.95 | 0.88 | 0.97 | 0.81 |

### D.2 Size of watermarked training data under fine-tuning scenario.

In this section, we shift our focus to a more realistic scenario where we fine-tune both the classifier and the watermarks using pre-trained models. We note that the new scenario differs from the one discussed in the main text, where the classifier is trained from scratch with randomly initialized weights.

To elaborate, we initially pretrain a set of watermarks along with their corresponding classifiers using a dataset such as MS-COCO. Subsequently, we fine-tune this pair using a new dataset. We present the results of this strategy by varying the number of new training data and summarize the outcomes in

Table 8 below. In contrast to the training from scratch scenario described in the main text, our RAW framework demonstrates a significant improvement with a reasonably small training dataset size.

Table 8: Summary of detection results under numbers of training data. AUROC (Ben) denotes the AUROC performance without image manipulations or adversarial attacks. AUROC (Adv) represents the average performance across nine distinct image manipulations and attacks.

| Dataset → | MS-COCO | | DBDiffusion | |
|---|---|---|---|---|
| | AUROC (Ben) ↑ | AUROC (Adv) ↑ | AUROC (Ben) ↑ | AUROC (Adv) ↑ |
| $n = 10$ | 0.74 | 0.68 | 0.70 | 0.68 |
| $n = 50$ | 0.85 | 0.71 | 0.86 | 0.72 |
| $n = 100$ | 0.99 | 0.85 | 0.99 | 0.81 |
| $n = 500$ | 0.99 | 0.91 | 0.99 | 0.93 |

### D.3 Different diffusion models for generating images

In this section, we examine the watermark detection results of our RAW approach across various generative models, specifically utilizing two widely recognized architectures: SDXL and BriXL. We maintain consistent settings as detailed in the main text. The AUROC outcomes are consolidated in Table 9. Our findings indicate that our RAW method achieves high AUROC scores for both MS-COCO and DBDiffusion datasets, underscoring its broad applicability.

Table 9: Summary of detection results under numbers of training data. AUROC (Ben) denotes the AUROC performance without image manipulations or adversarial attacks. AUROC (Adv) represents the average performance across nine distinct image manipulations and attacks.

| Dataset → | MS-COCO | | DBDiffusion | |
|---|---|---|---|---|
| | AUROC (Ben) ↑ | AUROC (Adv) ↑ | AUROC (Ben) ↑ | AUROC (Adv) ↑ |
| SDXL | 0.98 | 0.89 | 0.99 | 0.87 |
| BriXL | 0.99 | 0.91 | 0.99 | 0.89 |

### D.4 Additional Visual Examples

In this section, we present additional visual examples in Figure 5 and 4 resulting from our proposed RAW method. Our observations reveal no notably discernible differences between the original and watermarked images, highlighting the effectiveness of our approach.

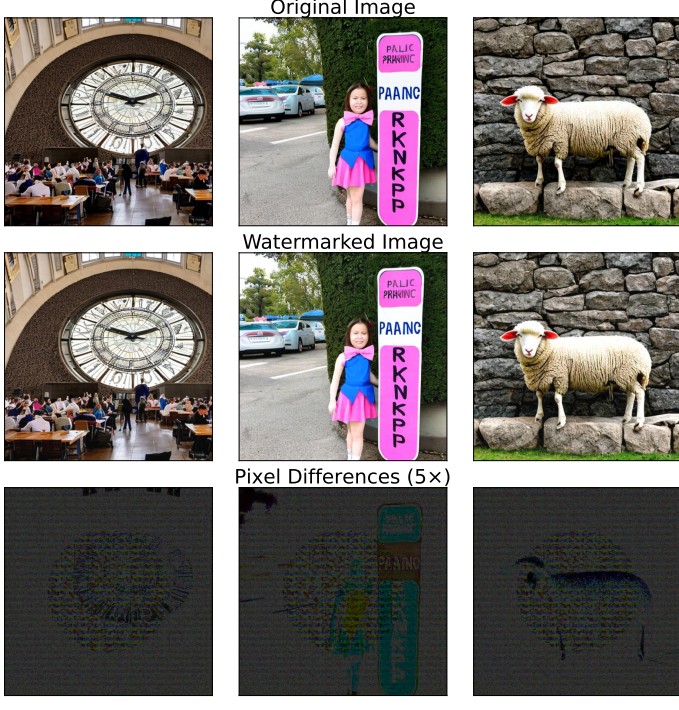

Figure 4: Examples of RAW-watermarked images (middle row)

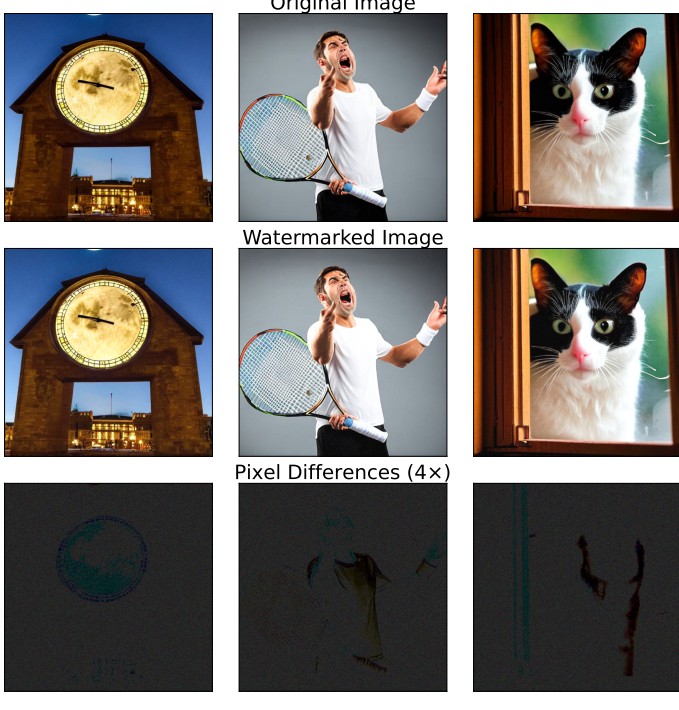

Figure 5: Examples of RAW-watermarked images (middle row)

