# OpenReview forum: "RAW: A Robust and Agile Plug-and-Play Watermark Framework for AI-Generated Images with Provable Guarantees"
_NeurIPS.cc/2024/Conference — NeurIPS 2024 poster_

### Official Review · Reviewer_FV9y · 2024-07-12

**Soundness:** 3
**Presentation:** 2
**Contribution:** 3
**Rating:** 5
**Confidence:** 4

**Summary:**

This paper introduces a novel image watermarking technique that injects signals into both the frequency and pixel domains of images. The watermark is specifically designed for the detection of AI-generated images and incorporates smoothing techniques to provide provable error guarantees against attacks under certain conditions. Additionally, the method's simplicity leads to faster watermarking speeds compared to existing techniques.

**Strengths:**

- Unlike most watermarking techniques in the literature which train an encoder decoder model without theoretical guarantees, this paper provides certified error guarantees against a certain range of attacks. This is a novel and valuable addition to the watermarking literature, since reliability is of great importance in this domain.

- While having a fast inference time, the watermark achieves high robustness against some common attacks.

**Weaknesses:**

- A comparison against more recent watermarking techniques is needed. In Table 4, the paper does not provide performance results (i.e., AUROC) against attacks compared to recent methods such as TreeRing [1], Trustmark [2], and RoSteLAS [3]. The included watermarking techniques (i.e., DwtDCT, RivaGAN) are known to be weaker against attacks compared to state-of-the-art watermarks.

- The paper lacks a comprehensive analysis of performance against image manipulations. While Table 4 illustrates performance on various attacks, it is unclear whether the hyper-parameters for these attacks have been set to their full strength. For instance, what hyper-parameters are being used for the VAE and Diff attacks? It is recommended that the authors include image quality metrics for the attacks (e.g., PSNR, SSIM) to demonstrate that their method maintains acceptable performance even under strong attacks that preserve image quality. Additionally, including a wider variety of diffusion attacks (e.g., pixel space and latent space) and potentially adversarial attacks (e.g., model substitution attacks), as discussed in the literature [4][5], would provide a more comprehensive evaluation.



[1] Yuxin Wen, John Kirchenbauer, Jonas Geiping, and Tom Goldstein. Tree-ring watermarks: Fingerprints for diffusion images that are invisible and robust, 2023.

[2] Tu Bui, Shruti Agarwal, and John Collomosse. Trustmark: Universal watermarking for arbitrary resolution images, 2023.

[3] Tu Bui, Shruti Agarwal, Ning Yu, and John Collomosse. Rosteals: Robust steganography using autoencoder latent space, 2023.

[4] Saberi, M., Sadasivan, V. S., Rezaei, K., Kumar, A., Chegini, A., Wang, W., and Feizi, S. Robustness of ai-image detectors: Fundamental limits and practical attacks, 2023.

[5] An, B., Ding, M., Rabbani, T., Agrawal, A., Xu, Y., Deng, C., Zhu, S., Mohamed, A., Wen, Y., Goldstein, T., et al.: Benchmarking the robustness of image watermarks, 2024.

**Questions:**

- Based on Table 3, RAW achieves a comparably low FID and high CLIP score, suggesting it applies an imperceptible and low-budget watermark signal to the images in the current evaluations. However, a more detailed evaluation of RAW's robustness against provable diffusion attacks [1][2] that claim to break imperceptible watermarks would be valuable. I am curious to know the authors' opinion on how their method might perform against such attacks.

[1] Saberi, M., Sadasivan, V. S., Rezaei, K., Kumar, A., Chegini, A., Wang, W., and Feizi, S. Robustness of ai-image detectors: Fundamental limits and practical attacks, 2023.

[2] Xuandong Zhao, Kexun Zhang, Zihao Su, Saastha Vasan, Ilya Grishchenko, Christopher Kruegel, Giovanni Vigna, Yu-Xiang Wang, and Lei Li. Invisible image watermarks are provably removable using generative ai, 2023.

**Limitations:**

- As noted in the limitations section, this technique is designed for use with a single watermark key (w) rather than multiple values, which is a disadvantage compared to existing methods. However, given the importance of the task of detecting AI-generated images, this focus on a single key value is justified and does not limit the value of this work.

---

> ### Author Rebuttal · Authors · 2024-08-07
>
> We sincerely thank the reviewer for devoting their valuable time to reviewing our paper and offering insightful suggestions for improving it.
>
> > Q: The paper lacks a comprehensive analysis of performance against image manipulations. While Table 4 illustrates performance on various attacks, it is unclear whether the hyper-parameters for these attacks have been set to their full strength. For instance, what hyper-parameters are being used for the VAE and Diff attacks?
>
> **R**: Thank you for your questions regarding the hyperparameters for the attacks. For all the attacks, we followed the public implementation in this GitHub repository (https://github.com/XuandongZhao/WatermarkAttacker) for watermark attacks without any modifications. We will include the hyperparameters for the attacks in the revision.
>
>
> > Q: It is recommended that the authors include image quality metrics for the attacks (e.g., PSNR, SSIM) to demonstrate that their method maintains acceptable performance even under strong attacks that preserve image quality.
>
> **R**: Thank you for your questions regarding the evaluation metrics.
>
> We report the PSNR and SSIM of the watermarked images in the Table below. For PSNR and SSIM, our method outperforms the StegaStamp method but is slightly behind DwtDctSvd and RivaGAN. These results are consistent with our expectations. The StegaStamp method achieves more robust detection by sacrificing image quality through injecting a large amount of noise into the image, while DwtDctSvd and RivaGAN maintain image quality slightly better at the cost of reduced robustness against (adversarial) image perturbations.
>
> | Method       | FID (Lower is preferred) | PSNR (Higher is preferred) | SSIM (Higher is preferred) |
> |--------------|---------------------------|----------------------------|----------------------------|
> | RAW          | 24.75                     | 31.9                       | 0.91                      |
> | StegaStamp   | 42.31                     | 27.8                       | 0.88                       |
> | DwtDctSvd    | 25.12                     | 37.6                       | 0.98                       |
> | RivaGAN      | 25.03                     | 35.2                       | 0.95                       |
>
>
> > Q: Additionally, including a wider variety of diffusion attacks (e.g., pixel space and latent space) and potentially adversarial attacks (e.g., model substitution attacks), as discussed in the literature [4][5], would provide a more comprehensive evaluation.
>
> **R**: Thank you for your suggestions regarding the evaluation of the proposed method. We will include more adversarial attacks, as you suggested, in the revision to provide a more comprehensive evaluation.

---

### Official Review · Reviewer_h31G · 2024-07-13

**Soundness:** 3
**Presentation:** 3
**Contribution:** 2
**Rating:** 5
**Confidence:** 5

**Summary:**

This paper proposes a Robust, Agile plug-and-play Watermarking (RAW) framework, which adds learnable watermarks directly on the original image and employs a classifier to detect the presence of the watermark. This design enhances both adaptability and computation efficiency, providing an model-agnostic for real-time applications. This paper show that RAW achieves provable guarantees on the false positive rate (FPR) for detection, even under adversarial attacks.

**Strengths:**

1. The proposed RAW provides provable guarantees on FPR for watermarking under signal processing and adversarial attacks.
2. RAW shows improved watermarking speed and robustness performance, while maintaining image quality.

**Weaknesses:**

1. In Equation (5), the watermark is also optimized, which indicates that this method can only add the same watermark. This limits the application.
2. The proposed method still embeds the watermark into the carrier image and is not specifically designed for AI-generated images.
3. In Table 2, some columns lack bolded data, and some bolded entries are not the best results.
4. The compared methods are outdated, failing to highlight the robustness and quality of this approach.
5. Since the carrier is involved in the method, PSNR should be evaluated.

**Questions:**

N/A

---

> ### Author Rebuttal · Authors · 2024-08-07
>
> We sincerely thank the reviewer for devoting their valuable time to reviewing our paper and offering insightful suggestions for improving it.
>
> > Q: The proposed method still embeds the watermark into the carrier image and is not specifically designed for AI-generated images.
>
> **R**: Thank you for your questions regarding the watermark design.
>
> The design of our proposed work aims to be usable by anyone, regardless of how the image is generated, whether by state-of-the-art diffusion models or drawn by an artist. Despite the general-purpose design, empirical evidence has verified the effectiveness of our method on AI-generated images. That being said, we will explore how to tailor the current methods to better fit AI-generated images in future work.
>
> > Q: In Table 2, some columns lack bolded data, and some bolded entries are not the best results.
>
> **R**: Thank you for your questions regarding the Table 2.
>
> We will bold all the best results for each column in the revision. In Table 2,there are a total of three performance metrics and two metrics regarding the quality of the watermarked images. For the performance of encoding speed, our method outperforms other methods by significant margins. For the normal AUROC, all methods are close to each other, and the differences are not significant. For the adversarial AUROC, our method achieved the second-highest AUROC of 0.92 for MS-COCO, slightly behind the StegaStamp method. This is reasonable and consistent with our expectations, as StegaStamp significantly degrades image quality to achieve robustness against image perturbations, as indicated by significantly higher FID and lower PSNR and SSIM values. We will make this clear in the revision.
>
> Table: Comparasion between StegaStamp and RAW (our method) on MS-COCO.
>
> | Method       | Adversarial AUROC |FID (Lower is preferred) | PSNR (Higher is preferred) | SSIM (Higher is preferred) |
> |--------------|---------------------------|---------------------------|----------------------------|----------------------------|
> Our method     | 0.92             | 24.75                     | 31.9                       | 0.91
> StegaStemp    | 0.93            | 42.31                     | 27.8                       | 0.88                      |
>
>
> > Q: Since the carrier is involved in the method, PSNR should be evaluated.
>
> **R:** Thank you for your suggestion regarding the image quality comparison metrics.
>
> We report the PSNR and SSIM of the watermarked images in the Table below. For PSNR and SSIM, our method outperforms the StegaStamp method but is slightly behind DwtDctSvd and RivaGAN. These results are consistent with our expectations. The StegaStamp method achieves more robust detection by sacrificing image quality through injecting a large amount of noise into the image, while DwtDctSvd and RivaGAN maintain image quality slightly better at the cost of reduced robustness against (adversarial) image perturbations.
>
> | Method       | FID (Lower is preferred) | PSNR (Higher is preferred) | SSIM (Higher is preferred) |
> |--------------|---------------------------|----------------------------|----------------------------|
> | RAW          | 24.75                     | 31.9                       | 0.91                      |
> | StegaStamp   | 42.31                     | 27.8                       | 0.88                       |
> | DwtDctSvd    | 25.12                     | 37.6                       | 0.98                       |
> | RivaGAN      | 25.03                     | 35.2                       | 0.95

---

> > ### Comment · Reviewer_h31G · 2024-08-12
> > **Final Comments**
> >
> > My concerns have not been well addressed, and even some have gone unanswered. I decide to lower the score and lean towards Borderline reject.

---

### Official Review · Reviewer_8Sds · 2024-07-17

**Soundness:** 3
**Presentation:** 3
**Contribution:** 2
**Rating:** 5
**Confidence:** 4

**Summary:**

This paper proposes a post-processing watermarking strategy that embeds watermarks into images after generation. The strategy is designed to be computationally efficient and model-agnostic. It involves training a learnable watermark and embedding it into both the frequency and spatial domains of the original image. The embedded watermark is later verified by a classifier, which is trained jointly with the watermark.

**Strengths:**

* Addresses an important problem.
* The problem formulation is clear and easy to understand.
* The paper is generally well-presented.
* Experimental results show improvements in adversarial robustness and speed.

**Weaknesses:**

* The survey of existing works in the "Related Works" section is incomplete. Numerous works on watermark frameworks for diffusion models are not included.
* It would be helpful if the authors provided a figure depicting an overview of the proposed method.
* The authors should clarify why they did not use signSGD to optimize the verification model if it is better. What are the data-level optimization problems?
* The reasoning for optimizing the watermark only based on $L_0$ instead of $L_{raw}$ needs to be explained.
* The method is not plug-and-play; it requires training the watermark and the verifier for each target generative model.
* Experimental evaluation:
  * For the SDXL model, no comparison with baselines is made.
  * Metrics for image quality comparison are limited. The authors should also include metrics like SSIM.

**Questions:**

see above.

**Limitations:**

see above.

---

> ### Author Rebuttal · Authors · 2024-08-07
>
> > Q: The survey of existing works in the "Related Works" section is incomplete. Numerous works on watermark frameworks for diffusion models are not included.
>
> **R**: Thank you for your questions regarding the related works section. We will include more works on watermark frameworks for diffusion models in the revision.
>
> > Q: It would be helpful if the authors provided a figure depicting an overview of the proposed method.
>
> **R**: Thank you for your suggestion regarding the presentation of the proposed method. We have included a figure depicting an overview of the proposed method in the revision.
>
> > Q: The authors should clarify why they did not use signSGD to optimize the verification model if it is better. What are the data-level optimization problems?
>
> **R**: Thank you for your insightful question regarding the optimizer and the data-level optimization problems.
>
> The data-level optimization problem refers to problems where the variables to be optimized are the data themselves instead of typical model parameters. For instance, the problem of finding an adversarial example is a data-level optimization problem ($x$ is an input to a machine learning model $f$, and $y$ is the correct label for $x$):
> $ min_{x'}  \| x - x' \|_\infty \text{;   subject to } f(x') \neq y .$ Data-level optimization problem can be difficult to solve, e.g., the results may get stuck at suboptimal points [A]. A common approach is to use the signum of first-order gradients [B,C].
>
> The reason for not using signSGD to optimize the verification model is that the verification model is a typical model parameter optimization problem, and signSGD may potentially lead to several optimization issues such as slow convergence speed. We will make this clear in the revision.
>
> Refs:
> [A] Wang et al., "Probabilistic margins for instance reweighting in adversarial training". In
> NeurIPS, 2021.
>
> [B] Madry et al., "Towards deep learning models resistant to adversarial attacks". In ICLR, 2018.
>
> [C] Wang et al., "Watermarking for Out-of-distribution Detection". In NeurIPS, 2022.
>
>
>
>
>
>
> > Q: The method is not plug-and-play; it requires training the watermark and the verifier for each target generative model.
>
> **R**: Thank you for your question regarding the plug-and-play nature of the proposed method. When referring to the method as plug-and-play, we mean that the watermarking method can be applied to any generative model without modification, regardless of its architecture. This is in sharp contrast to previous works that are designed for specific generative models, such as [8] for stable diffusion with the DDIM sampler only. In addition, we believe that additional training by individual users, e.g., artists, is actually beneficial since the watermark and verifier are tailored to them and possibly would bring better robustness. We will make this clear in the revision.
>
> > Q: The reasoning for optimizing the watermark only based on instead of needs to be explained.
>
> **R**: Thank you for your question regarding the optimization procedure.
>
> The main reason for optimizing the watermark based only on $L_0$ instead of $L_\text{raw}$ is because $L_\text{raw}$ involves non-differentiable terms with respect to the watermark, which makes gradient-based optimization infeasible. In detail, recall that $L_\text{raw}$ contains cross-entropy losses calculated on perturbed images, which can involve non-differentiable operations such as JPEG compression.
>
> > Q: Metrics for image quality comparison are limited. The authors should also include metrics like SSIM.
>
> **R:** Thank you for your suggestion regarding the image quality comparison metrics.
>
> We report the PSNR and SSIM of the watermarked images in the Table below. For PSNR and SSIM, our method outperforms the StegaStamp method but is slightly behind DwtDctSvd and RivaGAN. These results are consistent with our expectations. The StegaStamp method achieves more robust detection by sacrificing image quality through injecting a large amount of noise into the image, while DwtDctSvd and RivaGAN maintain image quality slightly better at the cost of reduced robustness against (adversarial) image perturbations.
>
> | Method       | FID (Lower is preferred) | PSNR (Higher is preferred) | SSIM (Higher is preferred) |
> |--------------|---------------------------|----------------------------|----------------------------|
> | RAW          | 24.75                     | 31.9                       | 0.91                      |
> | StegaStamp   | 42.31                     | 27.8                       | 0.88                       |
> | DwtDctSvd    | 25.12                     | 37.6                       | 0.98                       |
> | RivaGAN      | 25.03                     | 35.2                       | 0.95
>
> > Q:  For the SDXL model, no comparison with baselines is made.
>
> **R**: Thank you for your question regarding the baseline comparison for the SDXL model. We report the results in the following table. We observe that our proposed method outperforms RivaGAN and DwtDctSvd while being slightly behind StegaStamp. This is reasonable and consistent with our expectations, as explained previously.
>
> Table: Comparison between RAW and baselines on images generated by SDXL.
> | Method       | Average normal AUROC | Average Adversarial AUROC |FID (Lower is preferred) |
> |--------------|-------------------------------------|-----------------------------------------|-------------------------|
> | RAW          | 0.99                                | 0.90                                    | 15.3                   |
> | DwtDctSvd    | 0.99                                | 0.76                                    | 15.2                   |
> | RivaGAN      | 0.99                                | 0.80                                    | 15.6                 |
> | StegaStamp   | 0.99                                | 0.92                                    | 23.4                  |

---

> > ### Comment · Reviewer_8Sds · 2024-08-14
> > **Thanks for the response**
> >
> > The authors have addressed my concerns about the evaluation metric, SDXL model, and optimizer. Thanks! However, I still have concerns about the rest of the questions and consider them as weaknesses.  Hence, I decided to keep my score.

---

### Decision · Program_Chairs · 2024-09-25

**Decision:**

Accept (poster)

**Comment:**

3x BA. This paper proposes to embed watermarks into images after generation by adding learnable watermarks directly on the original image and employing a classifier to detect the presence of the watermark. The reviewers agree on the (1) important topic, (2) novel method, and (3) clear improvements. Most of the concerns, such as the insufficient evaluation, insufficient recent baselines, and insufficient analysis against image manipulations, have been addressed by the rebuttal. Therefore, the AC leans to accept this submission.